# Physical, sexual and psychological intimate partner violence and non-partner sexual violence against women and girls: a systematic review protocol for producing global, regional and country estimates

Heidi Stöckl [1], Lynnmarie Sardinha,[2] Mathieu Maheu-Giroux,[3] Sarah R Meyer [2], Claudia García-Moreno [2]

► Prepublication history and supplemental material for this paper is available online. To view these files, please visit the journal online (http://dx.doi.org/10.1136/bmjopen-2020-045574).

For numbered affiliations see end of article.

**Correspondence to**
Dr Claudia García-Moreno;
garciamorenoc@who.int

## ABSTRACT

**Introduction** In 2013, the WHO published the first global and regional estimates on physical and sexual intimate partner violence (IPV) and non-partner sexual violence (NPSV) based on a systematic review of population-based prevalence studies. In this protocol, we describe a new systematic review for the production of updated estimates for IPV and NPSV for global monitoring of violence against women, including providing the baseline for measuring Sustainable Development Goal to eliminate all forms of violence against women and girls.

**Methods and analysis** The systematic review will update and extend the previous search for population-based surveys (either nationally or subnationally representative) conducted among women aged 15+ years that measured the prevalence of physical, sexual, psychological and physical and/or sexual IPV, NPSV or sexual violence by any perpetrator up to December 2019. Data will be extracted separately for all age groups, setting (urban/rural), partnership status (currently partnered/ever partnered/all women) and recall period (lifetime prevalence/past 12 months). Studies will be identified from electronic searches of online databases of EMBASE, MEDLINE, Global Health and PsycInfo. A search of national statistics office homepages will be conducted for each country to identify reports on population-based, national or subnational studies that include data on IPV or NPSV published outside academic journals. Two reviewers will be involved in quality assessment and data extraction of the review. The review is planned to be updated on a continuous basis. All findings will undergo a country consultation process.

**Ethics and dissemination** Formal ethical approval is not required, as primary data will not be collected. This systematic review will provide a basis and a follow-up tool for global monitoring of the Sustainable Development Goal Target 5.2 on the elimination of all forms of violence against women and girls.

**PROSPERO registration number** CRD42017054100.

### Strength and limitations of this study

► Gold-standard systematic review process followed by country consultation process to verify global and country estimates on intimate partner violence (IPV).
► Multiple international technical advisory meetings to discuss the search procedures, data extraction and analysis plans of the estimates.
► Consensus on definitions and comparable measurements of physical and sexual IPV.
► Challenges in the comparability of data on non-partner sexual violence (NPSV) measurements due to a lack of global definitions and measurement tools.

## BACKGROUND

Intimate partner violence (IPV) and non-partner sexual violence (NPSV) are well recognised as human rights violations and public health problems.[1] In 2013, the WHO, the London School of Hygiene and Tropical Medicine, and the South African Medical Research Council conducted three systematic reviews to estimate the worldwide prevalence of IPV, NPSV and intimate partner homicide. They estimated that one in three women experienced physical and/or sexual IPV or NPSV in their lifetime; among all women murdered, 38.6% were murdered by their intimate partner.[2–5] The review further established associations between IPV and numerous adverse health outcomes, finding that women experiencing IPV were two times as likely to report depressive symptoms, nearly twice as likely to have alcohol use disorders, 16% more likely to have a low birthweight baby, and 1.5 times more likely to acquire HIV or another STI, compared

**BMJ**

with women who had not experienced IPV.[3] Similar associations were found for NPSV and depression and alcohol use disorders.[3 5]

The growing recognition of the high prevalence and significant health and other impacts of IPV and NPSV have contributed to the inclusion of 'the elimination of all forms of violence against women' in the 2030 Agenda for Sustainable Development. The Sustainable Development Goal (SDG) Target 5.2 on the elimination of all forms of violence against all women and girls in the public and private spheres is monitored using the following indicators:

> 5.2.1: Proportion of ever-partnered women and girls aged 15 years and older subjected to physical, sexual or psychological violence by a current or former intimate partner in the previous 12 months, by form of violence and by age.
>
> 5.2.2: Proportion of women and girls aged 15 years and older subjected to sexual violence by persons other than an intimate partner in the previous 12 months, by age and place of occurrence.

This article presents the protocol of the systematic review to generate the baseline data for producing estimates that will contribute to the global monitoring of violence against women, including for the SDG process.

## METHODS/DESIGN

This review builds on and updates the methodology used for the previous systematic reviews on the global prevalence of IPV and NPSV. It will be expanded to include evidence from studies directly received from national statistics offices, the Demographic and Health Surveys (DHS) and those identified through additional internet searches. A data extraction form and quality assessment form used for the 2013 prevalence review data extraction to collate data from selected studies will also be adapted and expanded to include psychological IPV, NPSV and sexual violence by any partner. The Preferred Reporting Items for Systematic Review and Meta-Analysis (PRISMA) guidelines are used as the template for reporting the present review. For the present protocol, the PRISMA statement for Protocols (PRISMA-P) was used for its reporting (see online supplemental appendix 1).[6] This review was registered in the PROSPERO International Prospective Register of systematic reviews on 2 January 2017.

### Aims of the review

To collect all available data on the prevalence of physical and/or sexual IPV, psychological violence by an intimate partner, NPSV and sexual violence by anyone, which can be used to measure advances in addressing violence against women and girls.

### Specific review questions

1. What is the prevalence of physical and/or sexual IPV in women 15 years and older, lifetime and in the last 12 months?
2. What is the prevalence of physical IPV only, lifetime and in the last 12 months?
3. What is the prevalence of sexual IPV only, lifetime and in the last 12 months?
4. What is the prevalence of psychological IPV, lifetime and in the last 12 months and how was it measured?
5. What is the prevalence of sexual violence by perpetrator(s) other than a partner only, lifetime and in the last 12 months?
6. What is the prevalence of sexual violence by any perpetrator, lifetime and in the last 12 months?
7. How does the prevalence of IPV and NPSV vary by age group, rural and urban settings over time, lifetime and in the last 12 months?

### Criteria for considering studies for this review
#### Inclusion

► Type of studies: nationally or subnationally representative population-based studies (cross-sectional or cohort studies).
► Type of participants: studies of women aged 15+.
► Type of outcome: studies measuring the prevalence of psychological, physical, sexual and physical and/or sexual IPV, NPSV and/or sexual violence by any perpetrator (or studies providing enough data to allow computing these estimates if not directly calculated or reported).
► Only acts based measures of psychological, physical and/or sexual IPV, non-partner violence and sexual violence by any partner will be included. The authors note that there is convergence on the definitions and standardised measures of physical and sexual IPV across the world. Psychological IPV and NPSV, however, are less well defined internationally and this systematic review therefore uses the authors' definition when extracting these data and notes the different definitions used and the diversity of acts covered to define them.

#### Exclusion

► Type of studies: case reports, case series, letters, reviews, policy reports, commentaries, editorials and administrative data including police statistics on reported crimes.
► Types of participants: studies in subgroups of participants that might not be generalisable to the whole population, such as clinical, school or prison samples or studies among pregnant women, same sex partners or police statistics of reported crimes.
► Outcome type: estimates that combine prevalence of physical and/or sexual IPV with other forms of violence, for example, psychological abuse or studies that did not use a 'gold-standard' measure of IPV defined here as measures asking about a range of

specific acts of experience of IPV because these avoid participants' subjective classification of experiences as 'violence' or not.

▶ Minimum reporting criteria: studies not giving sufficient detail for extrapolation or imputation, for example, concrete sample size, CIs or SE.

▶ Duplicate reports and/or publications of the same data: the less comprehensive/complete and up-to-date version will be excluded if the same data are reported on. Studies of different years or from different regions identified will all be captured.

▶ The results will only be extracted if they are representative for a single country, province or town or a geographically restricted area within a country.

### Search strategy for identifying relevant studies

A comprehensive and exhaustive search of the Ovid-based databases MEDLINE, EMBASE, Global Health and PsychInfo will be conducted on a regular basis, capturing the years 1989–2018 and thereafter to identify all relevant articles that contain data on the prevalence of psychological, physical and/or sexual IPV and NPSV and sexual violence by any perpetrator regardless of the language of publication. The search strategy of relevant terms is detailed in box 1. Both text words and medical subject heading terms will be used. This is a global study. Any study that is population based and fits the inclusion criteria will be included and no restrictions in terms of country names or languages will be applied.

### Searching of other data sources

Due to the increasing number of violence against women surveys conducted by local and national governments and the regular waves of the Demographic and Health Surveys (DHS) with a dedicated module on 'domestic violence' which includes IPV and NPSV, manual searches will also be conducted in DHS and other survey reports in a consistent manner in addition to signing up for the regular DHS updates.[7 8] We will search the webpages of governmental statistical and/or other offices of each individual country for reports that include data on the prevalence of IPV or NPSV. Each report will be traced back to its source to ensure they meet the inclusion criteria, for example, national representative survey on violence against women. Data would be excluded if it is administrative data, including police statistics on reported crimes, as these represent only the subset of few women who formally report violence and therefore are known to greatly underestimate the 'true' population prevalence.

### Selection of studies for inclusion in the review

Two independent reviewers will use Endnote X7 to independently identify articles and sequentially screen their titles and abstracts for eligibility. Full texts of articles deemed potentially eligible will be retrieved. Further, two independent reviewers will independently assess the full text of each study for eligibility, and consensually retain studies to be included. Dr García-Moreno will

---

**Box 1    Search terms and strategy for the Ovid-based databases MEDLINE, PsychInfo, EMBASE and Global Health**

1. meta.ab.
2. synthesis.ab.
3. literature.ab.
4. published.ab.
5. extraction.ab.
6. search.ab.
7. medline.ab.
8. selection.ab.
9. sources.ab.
10. trials.ab.
11. review.ab.
12. articles.ab.
13. .reviewed.ab.
14. english.ab.
15. language.ab.
16. randomized.hw.
17. trials.hw.
18. controlled.hw.
19. meta-analysis.pt.
20. review.pt.
21. or/1–20
22. epidemiologic studies/
23. exp case control studies/
24. exp cohort studies/
25. case control.tw.
26. (cohort adj (study or studies)).tw.
27. cohort analy$.tw.
28. (follow up adj (study or studies)).tw.
29. (observational adj (study or studies)).tw.
30. longitudinal.tw.
31. retrospective.tw.
32. cross sectional studies.tw.
33. cross sectional studies/
34. or/22–33
35. Animals/
36. Humans/
37. 35 not (35 and 36)
38. comment.pt.
39. letter.pt.
40. editorial.pt.
41. or/37–40
42. domestic violence/ or partner violence/ or spouse abuse/ or spouse violence/ or domestic abuse/ or partner abuse.mp. [mp=title, original title, abstract, name of substance word, subject heading word]
43. *battered women/
44. (intimate adj4 partner adj4 violence).tw.
45. (intimate adj4 partner adj4 abuse).tw.
46. (intimate adj4 partner adj4 victimi*).tw.
47. domestic abuse.tw.
48. spou$ abuse.tw.
49. dating violence.tw.
50. sexual abuse.tw.
51. [(partner or relationship or wom$n or domestic or spous*) adj4 (abus* or violen* or victimi* or batter*)].mp.
52. dating violence.tw.
53. sexual violence.tw.
54. rape.tw.
55. prevalence.tw.

Continued

---

---

**Box 1   Continued**

56. cross-sectional stud$.mp. [mp=title, original title, abstract, name of substance word, subject heading word]
57. survey.mp. [mp=title, original title, abstract, name of substance word, subject heading word]
58. health survey$.mp. [mp=title, original title, abstract, name of substance word, subject heading word]
59. or/42–54
60. or/55–58
61. 59 and 60
62. 61 not 41
63. 62 and (21 or 34)

---

resolve disagreements. A screening guide clearly stating the inclusion and exclusion criteria will be used to ensure that the selection criteria are reliably applied by two review authors. Eligible studies in languages other than English or French will be translated using native speakers or, if unavailable, Google Translate and considered for inclusion.

## Data extraction and management

Data will be extracted into an excel sheet independently by two individual researchers and divergences will be checked by a third extractor. In addition, 20% of the data entries will be checked by a third reviewer. Data extraction will furthermore be checked through cross tabulation of similar fields before data analysis. Data extraction fields will include:

1. Study identifier: study title, author and publication year.
2. Geographical information: region, country and iso3 codes.
3. Study characteristics: year of beginning of data collection and year of end of data collection, study design (cross-sectional, longitudinal), setting (urban, rural, subnational, national).
4. Population characteristics: denominator information (all women sampled, ever-partnered/ ever-married women or currently partnered/currently married women), and age range of the sample.
5. Information on violence underlying indicator: type of violence (physical, sexual, emotional/psychological, physical and/or sexual), type of violence by perpetrator (physical IPV, sexual IPV, psychological IPV, physical and/or sexual IPV, sexual violence non-partner, any sexual violence).
6. Who was asked about violence (all women vs ever partnered vs currently partnered), asked about IPV by current partner (current partner only or any partner), asked about violence by spouse (violence by spouse only vs violence by a spouse or partner), timing of violence (past year, past 2 years, ever), severe physical or sexual violence or not (definition by study).
7. Estimate: prevalence point estimate of specific form of violence, type of CI, lower and upper CI, SE, denominator, numerator.

8. Key quality indicators: study specified interviewer training on administering questions on violence against women, specialised violence against women survey or a module in a larger survey and whether study was national or subnational.

The quality of studies will be based on the following criteria, and these will be used as adjustment factors in the proposed analyses:

1. Whether the study was national or subnational, because prevalence may differ in areas within a country.
2. The type and time of violence measured (whether IPV was operationalised as physical only, sexual only, past year only, or severe only) because the prevalence of these forms of violence is lower than the combined lifetime experience of physical and/or sexual IPV.
3. Whether studies reported that interviewers were trained to ask about violence in a sensitive way as this increases participants willingness to disclose.
4. Whether studies were specifically designed to measure IPV or sexual violence prevalence.
5. If studies included only currently partnered women versus ever-partnered women, as estimates of IPV exposure in currently partnered women are likely to be lower than in ever-partnered women, particularly for lifetime prevalence, as they would not include women who have been in previously abusive relationships but not in a relationship at the time of the survey
6. If questions pertained to violence from the current partner and/or most recent partner only or from any partner.
7. If questions pertained to violence from a spouse only or from any intimate partner.

## Proposed data analysis

The data from this systematic review will be used to estimate the global, regional and national prevalence of physical and/or sexual IPV, in the past 12 months and lifetime for women aged 15 years and above. For NPSV and any sexual violence only global and regional estimates for lifetime prevalence will be produced.

For the preprocessing stage, in what we assume will be rare instances, where study's authors do not report information on the survey sample size, it will be estimated from the SE or CIs. If both are missing we will conservatively impute sample sizes corresponding to the lowest tercile of the overall distribution of survey samples sizes. If the overall survey sample size of a specific study is available but not the age-specific denominators of the prevalence estimates, we will impute them by distributing this overall sample size proportionally to the age-specific size of the female population reported in the United Nations World Population Prospect.[9]

For the analysis stage we will use a Bayesian meta-regression framework. This multilevel modelling approach will use survey-specific, country-specific and region-specific random effects to pool observations from different sources and improve accuracy of estimates by drawing on information from across units. The chosen

model structure is based on similar meta-regressions of health indicators.[2 5 10–17] Regions will be defined based on the Global Burden of Diseases classification. Regions will group countries in 21 mutually exclusive regions, which are situated in seven broad regions, based on the similarities of their epidemiological profiles.[3 18] Both nationally and subnationally representative studies will be included. We will assume for the latter that they provide estimates that could inherently be more variable than nationally representative studies. The advantage of the proposed multilevel modelling approach is that it will allow us to pool observations together from different sources and to 'borrow strength' across units. In case of a country with only one subnational survey with a small sample size, for example, an empirical observation from a similar country in the same region can improve its prevalence estimate's accuracy and precision. We will also use pooling, the sharing of information between observations to improve the calculation of global estimates. Through the use of multilevel models, this will be determined empirically by the data and not arbitrarily by the user.[19]

Additionally, this Bayesian regression model will consider heterogeneous age groups (using an age-standardising approach), account for country-specific age and time trends (using splines) and adjust for key survey differences (denominator, type of violence, etc—depending on the outcomes) through covariate modelling (with the adjustment calculated outside of the main model to avoid compositional bias). A joint model will be estimated for lifetime and past year IPV, imposing the constraint that all age-specific past year IPV estimates (for a particular country, at a particular time) should be lower or equal to those estimated for lifetime IPV.

### Country consultation process

Estimates of IPV generated in this review will undergo a country consultation process, as endorsed by the WHO Executive Board in 2001 through resolution (EB.107. R8) 'to ensure that each Member State is consulted on the best data to be used' for international estimation and reporting purposes.[20] In our particular case, the country consultation process was designed to enable countries to review information gathered from secondary data sources (ie, surveys/studies) that met the inclusion criteria; ensure the inclusion of any additional surveys/studies that meet these inclusion criteria, but which may not have been previously identified; and familiarise Member States with the statistical modelling approach used to derive the country, regional and global estimates. Focal points nominated by the countries received a summary of statistical methods, translated into all six United Nation official languages and their country profile for their critical review and feedback. The country profile outlined available data sources for the estimates, population-based surveys/studies that were excluded due to not meeting the inclusion criteria, covariates for adjustment, model fits and modelled national estimates. The country consultation process also allowed countries to suggest additional surveys/studies for review, provide previously missed data and reports from unpublished surveys/studies to be reviewed and to express their interest in conducting dedicated violence against women surveys in their countries.

### Patient and public involvement

No patient involved.

## ETHICS AND DISSEMINATION

Formal ethical approval is not required, as primary data will not be collected. This systematic review will provide a basis and a follow-up tool for global monitoring of the SDG Target 5.2 on the elimination of all forms of violence against women and girls.

## DISCUSSION

This systematic review will provide critical data for the global monitoring of progress to eliminate violence against women. It will help to track the effectiveness of efforts made by governments over the coming decade in preventing and addressing IPV and NPSV. The findings will directly feed into the global monitoring, including of the SDG Target 5.2 indicators.

Over the past years, governments have made commitments to increase the collection of data and research on violence against women, for example, in the WHO Global Plan of Action to strengthen the role of the health system in addressing violence, in particular against women and against children.[21] This includes conducting population-based studies to measure the prevalence and nature of violence against women, in particular IPV and sexual violence by non-partners. This is contributing to an expanding and robust evidence base that will allow tracking of changes over time. This systematic review will also contribute to the identification of challenges with the instruments and measures being used, especially for the forms of violence for which is there is less international agreement on measurement—psychological IPV and non-partner violence. It will also identify issues with how data is being reported.

The goal of this review and its future iterations is to contribute key epidemiological information on the magnitude and burden of violence faced by women around the world and how these prevalence patterns may differ by country, region, age and rural/urban contexts. These data should inform the development and implementation of effective policies and programmes to prevent and respond to violence against women.

The estimates based on the findings of this study will be used by governments, published in WHO/UN reports and as scientific papers in peer-reviewed journals and disseminated at relevant fora. The review will be updated regularly and the estimates will be updated approximately every 5 years to continue to monitor progress made in addressing violence against women.

## Review status

Data extraction is ongoing as several key data points are still missing, especially on NPSV. Existing extracted data on IPV are also currently undergoing a country consultation process, which is also adding new estimates. The methods for data analysis and for elaboration of the estimates have been tested and discussed by a technical advisory group (TAG) established by WHO and composed of independent experts. The TAG advises the UN Inter-Agency Working Group on Violence Against Women Data and Estimates convened by WHO.

#### Author affiliations
[1]Gender, Violence and Health Centre, Global Health and Development, LSHTM, London, UK
[2]The UNDP-UNFPA-UNICEF-WHO-World Bank Special Programme of Research, Development and Research Training in Human Reproduction (HRP), Department of Sexual and Reproductive Health and Research, World Health Organization, Geneva, Switzerland
[3]Epidemiology and Biostatistics, McGill University, Montreal, Quebec, Canada

**Acknowledgements** This database updates and builds on an earlier similar effort for the WHO, 2013 estimates on the prevalence of intimate partner violence and non-partner sexual violence which was led by Charlotte Watts at the LSHTM and Claudia García-Moreno of WHO. Karen Devries designed and oversaw the earlier systematic searches for intimate partner violence and Naeema Abrahams of the South African Medical Research Council for non-partner sexual violence.

**Contributors** HS adapted the original search strategy and study design to align it with the discussions on global monitoring of Violence Against Women, including measurement of the indicators related to Sustainable Development Goal Target 5.2. and developed the inclusion of psychological abuse and any sexual violence into the search and analysis, expanded the search strategy to capture data and reports outside the peer-reviewed literature, participated in deciding on the analysis strategy and drafted the current manuscript. LS inputted into the adapted search strategy, study design and data extraction sheet, refined inclusion and exclusion criteria and quality indicators and has been involved in all decisions regarding the current search protocol and this manuscript. MM-G contributed to the establishment of the list of key covariates and study characteristics to collect, methods to check the database for integrity and consistency, and conceived the data analysis plan. SRM inputted into the adapted search strategy, study design and data extraction sheet, refined inclusion and exclusion criteria and quality indicators and has been involved in all decisions regarding the current search protocol and this manuscript. CG-M co-conceived the original search strategy and study design, inputted into the data extraction sheet, inclusion and exclusion criteria and quality indicators, has been involved in all decisions regarding the current search protocol and this manuscript and provides technical oversight to the whole project. She is the guarantor of the review. All authors read and approved the final manuscript.

**Funding** This work received funding from the Department for international Development, through the UN Women-WHO Joint Programme on Strengthening Violence Against Women Measurment and Data Collection and the UNDP-UNFPA-UNICEF-WHO-World Bank Special Programme of Research, Development and Research Training in Human Reproduction (HRP), a cosponsored programme executed by the WHO.

**Disclaimer** The funders had no role in the design of the review. Grand number: Not applicable.

**Competing interests** None declared.

**Patient and public involvement** Patients and/or the public were not involved in the design, or conduct, or reporting, or dissemination plans of this research.

**Patient consent for publication** Not required.

**Provenance and peer review** Not commissioned; externally peer reviewed.

**ORCID iDs**
Heidi Stöckl http://orcid.org/0000-0002-0907-8483
Sarah R Meyer http://orcid.org/0000-0001-8595-2358
Claudia García-Moreno http://orcid.org/0000-0002-0208-4119

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
