## [Reviewer comments · BMJ Open]

ARTICLE DETAILS

TITLE (PROVISIONAL)	Physical, sexual and psychological intimate partner violence and non-partner sexual violence against women and girls: a systematic review protocol for producing global, regional and country estimates
AUTHORS	Stöckl, Heidi; Sardinha, Lynnmarie; Maheu-Giroux, Mathieu; Meyer, Sarah; Garcia-Moreno, Claudia

VERSION 1 – REVIEW

REVIEWER	Taft, Angela La Trobe University, Mother and Child Health Research
REVIEW RETURNED	18-Nov-2020

GENERAL COMMENTS	I suggest that a few more references would be helpful in this review. 1. First the website for the DHS repositories, second for your meta-regression analyses and third GBD classification. 2. Consider further categorisation of the outcomes by low, middle and high income countries disparities.
--

REVIEWER	Fisher, Jane Monash University, Jean Hailes Research Unit, School of Public Health and Preventative Medicine
REVIEW RETURNED	29-Nov-2020

GENERAL COMMENTS	Violence against women and girls is of serious international concern and, despite efforts to reduce it, persists as a severe problem experienced in all countries and settings. This protocol for a systematic review is very clearly written, with a persuasive rationale and explanation of what it will contribute beyond what is already known. The methods are appropriate and for the most part replicable. My questions are relatively minor: 1. While the search of the literature and the approach to data extraction and quality assessment are clearly described, the process for undertaking the country consultations was not. Please add a description of who will collect these data, how informants will be identified and recruited, what kind of interview will be conducted, and how data will be recorded and analysed. 2. Please explain why the quality of studies will be assessed using study-specific criteria rather than one of the standard quality assessment systems. If the study-specific criteria are retained, how will these be scored?
--

	3. Please describe how you will report on countries for which there are no data available?
--	--

VERSION 1 – AUTHOR RESPONSE

Reviewer: 1 Dr. Angela Taft, La Trobe University

Comments to the Author:

I suggest that a few more references would be helpful in this review.

1. First the website for the DHS repositories, second for your meta-regression analyses and third GBD classification.

Thank you. We have included additional references as you have mentioned.

2. Consider further categorisation of the outcomes by low, middle and high income countries disparities.

This is an excellent suggestion. So far, we planned to produce global, regional and country estimates, according to the global burden of disease regions and super-regions. We will also consider creating estimates by different low, middle and high income country categories if separate regional categorizations are required by the WHO or the other members of the UN partners.

Reviewer: 2. Prof. Jane Fisher, Monash University

Comments to the Author:

Violence against women and girls is of serious international concern and, despite efforts to reduce it, persists as a severe problem experienced in all countries and settings. This protocol for a systematic review is very clearly written, with a persuasive rationale and explanation of what it will contribute beyond what is already known. The methods are appropriate and for the most part replicable. My questions are relatively minor:

1. While the search of the literature and the approach to data extraction and quality assessment are clearly described, the process for undertaking the country consultations was not. Please add a description of who will collect these data, how informants will be identified and recruited, what kind of interview will be conducted, and how data will be recorded and analysed.

Thank you for pointing out the need for further explanation on this point. This has been added after the data analysis section as a separate section of the submitted manuscript.

2. Please explain why the quality of studies will be assessed using study-specific criteria rather than one of the standard quality assessment systems. If the study-specific criteria are retained, how will these be scored?

The team involved in this systematic review has been quite fortunate to be able to draw on the prior experience in conducting systematic reviews for the global burden of disease study on the global prevalence of physical and or sexual intimate partner violence and non-partner sexual violence as well as global prevalence of intimate partner homicide. We therefore knew that the classic systematic review process of examining published peer-reviewed journal article only yields approximately a third of the needed data, while the other third could be retrieved from the Demographic and Health Surveys and other major surveys as well as published, yet not peer-reviewed reports on national prevalence surveys. Various standard quality assessment systems, e.g. AXIS[1] were considered by the Global Technical Expert group early on. Yet, knowing the nature of the expected data and studies, they were not deemed useful given their lack of appropriateness to judge whether estimates are reliably measuring intimate partner violence or non-partner sexual violence. It was decided that whether a study was focused on violence against women and girls, and whether interviewers were specifically trained on researching violence was a more meaningful study quality criteria than criteria that generally describe cross-sectional studies e.g. response rate or the survey mode. The study specific criteria were included in the data modelling, with more weight given to

studies with a stronger design.

3. Please describe how you will report on countries for which there are no data available?

We have added information on this in the study protocol to further explain our multilevel modelling technique:

The chosen model structure is based on similar meta-regressions of health indicators [2-11]. Regions will be defined based on the Global Burden of Diseases (GBD) classification. Regions will group countries in 21 mutually exclusive regions, which are situated in seven broad regions, based on the similarities of their epidemiological profiles [12, 13]. Both nationally and sub-nationally representative studies will be included. We will assume for the latter that they provide estimates that could inherently be more variable than nationally representative studies. The advantage of the proposed multilevel modelling approach is that it will allow us to pool observations together from different sources and for the model to “borrow strength” across units. In case of a country with only one sub-national survey with a small sample size, for example, an empirical observation from a similar country in the same region can improve its prevalence estimate’s accuracy and precision. We will also use pooling, the sharing of information between observations to improve the calculation of global estimates. Through the use of multilevel models, this will be determined empirically by the data and not arbitrarily by the user [14].